# Advances in Molecular Understanding of Polycythemia Vera, Essential Thrombocythemia, and Primary Myelofibrosis: Towards Precision Medicine

**DOI:** 10.3390/cancers16091679

**Published:** 2024-04-26

**Authors:** Hammad Tashkandi, Ismail Elbaz Younes

**Affiliations:** 1Department of Pathology and Laboratory Medicine, Moffitt Cancer Center, Tampa, FL 33612, USA; 2Department of Laboratory Medicine and Pathology, Division of Hematopathology, University of Minnesota, Minneapolis, MN 55455, USA; elbaz008@umn.edu

**Keywords:** myeloproliferative neoplasms, polycythemia vera, essential thrombocythemia, primary myelofibrosis, molecular diagnostics, next-generation sequencing, targeted therapies, variant allele frequencies, clonal evolution

## Abstract

**Simple Summary:**

Myeloproliferative neoplasms (MPNs) represent a group of blood cancers characterized by the excessive production of blood cells in the bone marrow, including Polycythemia Vera (PV), Essential Thrombocythemia (ET), and Primary Myelofibrosis (PMF). Recent advancements in the field of molecular biology have significantly enhanced our understanding of the genetic underpinnings of these conditions. The identification of specific genetic mutations has refined the accuracy of diagnoses and paved the way for personalized therapeutic approaches. By tailoring treatment strategies to the individual genetic profile of a patient’s disease, clinicians can optimize clinical outcomes and improve the quality of life for those affected. This summary aims to elucidate the recent molecular discoveries in PV, ET, and PMF, highlighting their pivotal role in the evolution of patient management strategies.

**Abstract:**

Myeloproliferative neoplasms (MPNs), including Polycythemia Vera (PV), Essential Thrombocythemia (ET), and Primary Myelofibrosis (PMF), are characterized by the clonal proliferation of hematopoietic stem cells leading to an overproduction of hematopoietic cells. The last two decades have seen significant advances in our understanding of the molecular pathogenesis of these diseases, with the discovery of key mutations in the *JAK2*, *CALR*, and *MPL* genes being pivotal. This review provides a comprehensive update on the molecular landscape of PV, ET, and PMF, highlighting the diagnostic, prognostic, and therapeutic implications of these genetic findings. We delve into the challenges of diagnosing and treating patients with prognostic mutations, clonal evolution, and the impact of emerging technologies like next-generation sequencing and single-cell genomics on the field. The future of MPN management lies in leveraging these molecular insights to develop personalized treatment strategies, aiming for precision medicine that optimizes outcomes for patients. This article synthesizes current knowledge on molecular diagnostics in MPNs, underscoring the critical role of genetic profiling in enhancing patient care and pointing towards future research directions that promise to further refine our approach to these complex disorders.

## 1. Introduction

Myeloproliferative neoplasms (MPNs), including polycythemia vera (PV), essential thrombocythemia (ET), and primary myelofibrosis (PMF), constitute a group of hematologic malignancies characterized by clonal expansion of one or more myeloid lineages, which results in excessive production of hematopoietic cells. Traditionally, the diagnostic framework for these conditions has predominantly relied on morphological and clinical parameters. Nevertheless, the past two decades have witnessed a significant shift in our comprehension of the genetic underpinnings of these disorders, prompted by the discovery of the *JAK2* V617F mutation and subsequent identification of mutations in *CALR* and *MPL*. These findings have fundamentally altered the diagnostic and treatment paradigms for PV, ET, and PMF [1,2].

The fifth edition of the World Health Organization (WHO) Classification of Hematolymphoid Tumors and the International Consensus Classification (ICC) have reaffirmed the central role of molecular diagnostics in the classification and management of MPNs [3,4]. While the diagnostic criteria for PV, ET, and PMF have seen no significant changes in the light of these updates, the advancements in our understanding of their molecular pathology have been profound. The recognition of additional somatic mutations beyond the driver mutations in *JAK2*, *CALR*, and *MPL* has deepened our insight into the disease biology, offering new avenues for targeted therapies and prognostic assessments [5,6].

The aim of this review is not to provide a comprehensive examination of each entity within the spectrum of MPNs, where there exists a lack of complete concordance between the WHO and the ICC; rather, it is to underscore the critical importance of molecular diagnostics in the clinical management of PV, ET, and PMF, against the backdrop of the broader spectrum of MPNs. It offers a concise exploration of key molecular markers and elucidates their influence on diagnosis, prognosis, and therapeutic approaches. Furthermore, this review includes a forward-looking discussion on future directions, aiming to provide insights into potential advancements and the evolving intersection of genetic research with clinical practice in the MPN domain.

## 2. Diagnostic Criteria for PV, ET, PMF, and MPN-NOS

In alignment with the fourth edition of the WHO [7], the recent updates to the WHO and ICC 2022 diagnostic criteria continue to incorporate both molecular and clinical features, reaffirming their crucial role in the management and therapeutic strategies for MPNs. Although these modifications are minimal, they emphasize the pivotal role of molecular diagnostics in contemporary hematological practice. Table 1 outlines the diagnostic criteria for PV, ET, PMF, and Myeloproliferative Neoplasm, Not Otherwise Specified (MPN-NOS), also known as Myeloproliferative Neoplasm, Unclassifiable (MPN-U) in ICC criteria [3,4].

MPN-NOS/MPN-U is particularly pertinent for patients in the preliminary phases of the disease, where diagnostic characteristics are not fully manifest, and requisite thresholds for a definitive diagnosis are not achieved. These cases require ongoing monitoring to ascertain the emergence of a specific MPN subtype as the disease progresses. This classification is also applicable to individuals presenting with splanchnic or portal vein thrombosis who do not satisfy the diagnostic criteria for a particular MPN subtype [4]. Notably, MPN-NOS/MPN-U may include instances where molecular indications of myeloproliferation are present, yet concurrent neoplastic or inflammatory conditions obscure the typical morphological diagnostic indicators. 

For cases displaying significant morphologic dysplasia or absolute monocytosis at diagnosis, the integration of molecular data with comprehensive bone marrow evaluations is crucial. These assessments help to differentiate such cases from Myelodysplastic Syndromes (MDS) or MDS/MPN overlap syndromes, and from the advanced stages of MPN during disease progression.

## 3. Advances in Molecular Understanding of MPNs: Impacts on Diagnosis, Prognosis, and Treatment

The integration of next-generation sequencing (NGS) into the diagnostic regimen for MPNs has uncovered a complex landscape of genetic mutations, necessitating an advanced framework for interpretation that categorizes these mutations into diagnostic, prognostic, and therapeutic groups. Such classification is vital for patient-specific management but is challenged by overlapping genetic markers across MPN subtypes and the advent of novel mutations. The identification of key driver mutations in genes like *JAK2*, *CALR*, and *MPL* is crucial for the diagnosis of PV, ET, and PMF and forms the basis of disease classification according to WHO and ICC guidelines [3,4]. Furthermore, the detection of secondary mutations contributes to our understanding of disease progression and variability in outcomes, highlighting the role of comprehensive genetic profiling in refining risk assessment and guiding treatment decisions [8,9,10]. On the therapeutic front, the identification of actionable mutations has led to the development of tailored treatment approaches, exemplified by the transformative impact of JAK inhibitors on MPN management. This correlation between genetic insights and therapeutic efficacy accentuates the need for ongoing molecular monitoring to enhance treatment responses and address the development of drug resistance [11]. This review also integrates the nomenclature proposed by Luque Paz et al., distinguishing between clonal and disease drivers in MPNs, enriching the discourse on the genetic underpinnings of these diseases [12]. Table 2 provides a summary of the genetic mutations present in MPNs.

### 3.1. Disease Drivers

The discovery of the *JAK2* V617F mutation in a significant majority of MPN patients marked a seminal moment, providing a molecular hallmark for PV, ET, and PMF [25,26]. This mutation results in constitutive activation of the JAK-STAT signaling pathway, driving the uncontrolled proliferation characteristic of MPNs. Following this discovery, mutations in *CALR* and *MPL* were identified [14,27,28,29], adding layers to the molecular diagnostic landscape. *CALR* mutations are found predominantly in ET and PMF patients lacking the *JAK2* mutation, while *MPL* mutations occur less frequently in MPNs. The detection of these mutations influences disease mechanisms and phenotypes, offering a basis for confirming MPN diagnoses, classifying patients into specific prognostic categories, and informing targeted treatment strategies.

#### 3.1.1. Implications of *JAK2* Mutations in MPNs

The *JAK2* V617F mutation in exon 14, present in approximately 96% of PV cases, 55% of ET cases, and 65% of PMF cases [2], is associated with abnormal hematopoiesis and enhanced sensitivity of hematopoietic stem and progenitor cells to growth factors [30]. Conversely, mutations in *JAK2* exon 12, mainly associated with PV, result in increased erythroid cell production [31,32]. Additionally, less common insertions and deletions within exon 12 of *JAK2*, occurring in 2% to 3% of PV patients, lead to increased erythroid proliferation [33,34]. These mutations are essential for diagnosing MPNs, influencing both prognosis and the selection of therapeutic strategies. The disruption of the JAK-STAT signaling pathway by the *JAK2* V617F mutation highlights the utility of JAK2 inhibitors like fedratinib or ruxolitinib in treating symptomatic or advanced stages of MPNs [6]. The role of variant allele frequency (VAF) in this context will be discussed elsewhere in this review. 

#### 3.1.2. Implications of CALR Mutations in MPNs

Calreticulin (*CALR*) mutations, central to the pathology of MPNs, notably influence ET and PMF. These mutations, primarily categorized into Type 1, featuring a 52 bp deletion disrupting calcium binding and activating the IRE1α/XBP1 unfolded protein response pathway [35], and Type 2, a 5 bp insertion causing frameshift and novel protein C-termini, account for 20–30% of ET and 25–35% of PMF cases, respectively. Overall, they represent 60–80% of patients with *JAK2/MPL*-negative ET and PMF [14,27]. Additionally, Type 1-like and Type 2-like mutations extend the *CALR* mutation spectrum, sharing structural similarities that lead to analogous frameshifts and novel C-terminal peptides. *CALR* mutations are linked to better overall survival than *JAK2* V617F or *MPL* W515 mutations, with this advantage more pronounced in those with Type 1/Type 1-like mutations [13,17,36]. In PMF, *CALR* mutations exhibit prognostic superiority for better overall survival in comparison to *JAK2* V617F mutations or triple-negative status [13], even when adjusting for age in multivariate analyses. The cumulative incidence of leukemic transformation is lower in *CALR*-mutated patients than in those with *JAK2* or *MPL* mutations or triple-negative disease. The prognosis varies between the two *CALR* mutation types; patients with Type 1/Type 1-like mutations have significantly longer survival than those with Type 2/Type 2-like mutations or *JAK2* V617F mutations, and a lower rate of leukemic transformation compared to Type 2/Type 2-like mutations [17]. 

#### 3.1.3. Implications of MPL Mutations in MPNs

Activating mutations in the thrombopoietin receptor gene (*MPL* W515L/K) occur in about 5% to 8% of PMF patients and 1% to 4% of ET patients [28,29,37]. These *MPL* mutations mark a distinct pathogenetic group within MPNs, significantly affecting the disease progression of ET and PMF. These mutations, while less common than those in *JAK2* and *CALR*, are driving myeloproliferation in the absence of the latter mutations, significantly influencing the progression of ET and PMF. In PMF patients, *MPL* mutations correlate with lower hemoglobin levels at diagnosis and a higher need for transfusion, underscoring their impact on disease severity [38].

Patients lacking all three primary driver mutations—*JAK2*, *CALR*, or *MPL*, known as having a triple-negative mutation status—comprise about 10% of the MPNs and are associated with a poorer prognosis [15,39].

### 3.2. Clonal Drivers

Clonal driver mutations, in isolation, do not induce the MPN phenotype within murine models [12]. However, when these mutations coexist alongside a “disease driver” mutation, they exhibit the capacity to alter the phenotypic manifestation of the disease and clinical outcome.

#### 3.2.1. Understanding Clonal Evolution and the Role of Additional Mutations in MPNs

A study examining 22 MPN patients with the *JAK2* V617F mutation and additional somatic mutations underscored the prognostic value of clonal architecture, identifying four distinct clusters, with one linked to decreased survival across MPN subtypes [40]. This highlights the role of clonal evolution, driven by initial mutations in *JAK2*, *CALR*, or *MPL* and further complicated by mutations affecting epigenetic regulation (e.g., *ASXL1*, *TET2*), RNA splicing (e.g., *SRSF2*, *U2AF1*), and signaling pathways, in disease progression, treatment resistance, and the development of clonal heterogeneity. This heterogeneity, revealed through advancements in single-cell genomics, shows multiple subclones within a single patient, impacting disease behavior and treatment outcomes.

The discovery of additional mutations beyond *JAK2*, *CALR*, and *MPL* elucidates the complexity of MPNs, contributing to clinical heterogeneity and diverse therapeutic responses. High-molecular-risk (HMR) mutations, such as *ASXL1*, indicate a more severe disease course [6], while the sequential acquisition of *JAK2* and *TET2* mutations influences the clinical phenotype [41]. Advanced sequencing technologies facilitate the detection of subclonal mutations, enabling refined patient stratification into risk categories and the formulation of personalized treatment strategies based on the genetic profile of the disease. This comprehensive approach underscores the necessity for personalized medicine in MPNs, addressing the full spectrum of genetic diversity to overcome treatment resistance and improve patient management.

#### 3.2.2. Mutation Profiles as Prognostic Markers

The mutation spectrum in MPNs serves as a critical determinant of disease heterogeneity, influencing disease progression and treatment efficacy. Incorporating comprehensive mutation profiles into prognostic models like the Myelofibrosis Prognostic Scoring System (MIPSS) and the Genetically Inspired Prognostic Scoring System (GIPSS) enhances risk stratification precision. The MIPSS integrates genetic and clinical factors to predict survival in PMF [42,43], including key mutations and clinical parameters. The GIPSS, focusing solely on genetic aberrations, stratifies risk in myelofibrosis patients based on the presence of specific genetic markers [44,45].

#### Myelofibrosis Prognostic Scoring System (MIPSS)

The MIPSS is a comprehensive scoring system that integrates both genetic and clinical factors to predict survival outcomes in patients with PMF. It was developed in response to the limitations of traditional scoring systems (IPSS) that relied primarily on clinical features. MIPSS includes mutations in key genes known to influence disease progression, such as *JAK2*, *CALR*, and *MPL*, along with additional mutations in genes like *ASXL1*, *EZH2*, *SRSF2*, and *IDH1/2*, which have been identified as high-molecular-risk (HMR) mutations due to their association with adverse outcomes. Clinical factors such as age, hemoglobin level, white blood cell count, and the presence of constitutional symptoms are also considered in the scoring [1,20,42].

Subsequent research on PV involving 336 patients identified specific risk factors, including the *SRSF2* mutation, age over 67, elevated leukocyte counts, and thrombosis history, leading to the creation of the MIPSS-PV [1]. This variant stratifies PV patients into three risk categories—low, intermediate, and high—with median overall survival (OS) rates differing significantly across groups. Similarly, an analysis of 451 ET patients highlighted the influence of specific mutations (*SF3B1*, *SRSF2*, *TP53*, *U2AF1*), age above 60, male gender, and high leukocyte counts on survival, culminating in the development of the MIPSS-ET [1]. This model also segments ET patients into three risk levels, each with distinct median OS projections. These findings underscore the necessity for further validation to solidify these prognostic tools’ roles in clinical practice.

#### Genetically Inspired Prognostic Scoring System (GIPSS)

The GIPSS focuses exclusively on genetic aberrations to stratify risk in patients with myelofibrosis. This approach is based on the premise that genetic mutations are strong independent predictors of survival and leukemic transformation in myelofibrosis. The GIPSS categorizes patients into risk groups (low, intermediate, and high) based solely on the presence or absence of specific genetic markers, including driver mutations (*JAK2*, *CALR*, *MPL*) and the aforementioned HMR mutations. The advantage of the GIPSS lies in its simplicity and the direct implication of genetic abnormalities in determining prognosis, offering a clear framework for risk stratification that can be particularly useful in guiding therapeutic decisions and managing patient expectations [44,45].

As research in the field progresses and new genetic markers are identified, it is expected that these prognostic models will continue to evolve, further enhancing their utility in clinical practice.

### 3.3. Molecular Profile and Prognostic Implications in Blast Phase MPNs

The molecular profile of the blast phase in MPNs (MPN-BP) plays a crucial role in determining patient outcomes, highlighting the intricate genetic alterations that drive leukemic progression and resistance to therapy. Research involving 248 patients who developed MPN-BP that used targeted next-generation sequencing (NGS) revealed a complex mutational spectrum. The prevalence of primary driver mutations was found to be *JAK2* (57%), *CALR* (20%), and *MPL* (9%), with 13% being triple-negative, and 85% of cases exhibited additional mutations. Among these, mutations in *ASXL1* (47%), *TET2* (19%), *RUNX1* (17%), *TP53* (16%), *EZH2* (15%), and *SRSF2* (13%) were significant, suggesting their contributory role in MPN-BP pathogenesis. *RUNX1* mutations, in particular, emerged as independent indicators of poorer survival, highlighting their critical prognostic value irrespective of the therapeutic strategy. This study identifies mutations such as *RUNX1*, *TP53*, *EZH2*, and *PTPN11* as potentially central to the leukemic transformation in MPN, with *RUNX1* mutations notably impacting survival adversely [23]. Supplementing these findings, a more recent study differentiates MPN-accelerated phase (AP)/BP from de novo AML by a higher prevalence of *TP53* and *IDH1/2* mutations and fewer *FLT3* and *DNMT3A* mutations. Moreover, it underscores the association between leukemic transition and several risk factors, including age, anemia, and specific genetic markers [46].

### 3.4. Understanding Triple-Negative MPNs: Diagnostic and Molecular Challenges

MPNs without *JAK2*, *CALR*, and *MPL* mutations, known as triple-negative MPNs (TN-MPNs), account for about 9–15% of all MPN cases and pose significant diagnostic and therapeutic challenges due to the absence of conventional molecular markers [15,47]. The diagnostic ambiguity surrounding TN-MPNs underscores the importance of extensive molecular profiling, with next-generation sequencing (NGS) panels crucial for identifying a broader array of genetic alterations that could assist in patient management. Research into the genetic underpinnings of TN-MPNs has identified a range of noncanonical gene mutations involved in epigenetic regulation, RNA splicing, and signaling pathways, which are instrumental in understanding the disease’s pathogenesis and prognostication [48].

TN primary myelofibrosis (TN-PMF) demonstrates unique clinicopathologic and molecular characteristics, often presenting with thrombocytopenia, absence of organomegaly, specific bone marrow features like diminished granulocytic elements and dyserythropoiesis, cytogenetic anomalies such as trisomy 8, and a higher frequency of *ASXL1/SRSF2* co-mutations. Such distinctive features necessitate tailored diagnostic and therapeutic approaches for TN-PMF patients [49]. 

Prognostically, TN-MPN patients face poorer outcomes, with studies indicating a median overall survival (OS) of only 3 years for triple-negative PMF, compared to 18 years for those with *CALR* mutations and 9 years for those with *JAK2* V617F mutation or *MPL* mutation [13]. This stark contrast in survival rates, coupled with a higher incidence of leukemic transformation in triple-negative cases [13,16], emphasizes the critical need for improved management strategies for this subgroup. 

### 3.5. Challenges Introduced by Clonal Hematopoiesis of Indeterminate Potential 

Clonal Hematopoiesis of Indeterminate Potential (CHIP) is recognized for its role in the intersection of aging, myelodysplastic neoplasms (MDS), MPNs, and cardiovascular diseases [50,51,52]. The *JAK2* V617F mutation, a primary driver in MPN pathogenesis, is notably linked to CHIP [53], highlighting the intricate relationship between these conditions. In CHIP instances, the *JAK2* V617F mutation may emerge as the sole genetic alteration [50], underlining its potential as an early initiator of MPNs and its association with increased risks of coronary heart disease [51] and venous thrombosis [54]. The distinction between early-stage MPNs and CHIP presents a clinical challenge, especially given the capacity of *JAK2*, *CALR*, or *MPL* mutations to produce MPN-like phenotypes in animal studies [55]. This underscores the need for strategies to evaluate the progression likelihood of CHIP to MPNs, enhancing patient management through potentially more intensive monitoring for those at higher risk [56]. 

Furthermore, the *JAK2* V617F mutation activates kinase and abnormal signaling pathways, with CHIP-associated mutations like *DNMT3A* and *TET2* fostering a subclinical inflammatory state that could pave the way for pathological developments, including infections. This situation highlights the pro-inflammatory role of CHIP-related mutations in aging and their contribution to inflammation, coagulation, thrombosis, and eventual progression to overt hematologic malignancies [57].

CHIP’s presence, especially with MPN-associated mutations, can precede and potentially trigger chromosomal abnormalities, increasing lymphoid and myeloid malignancies’ risk. The co-occurrence of CHIP mutations with chromosomal aberrations indicates that CHIP often heralds the arrival of chromosomal anomalies, signifying a crucial phase in MPN development [58].

Highly sensitive polymerase chain reaction (PCR) techniques have identified *JAK2* V617F VAFs as low as 0.01%, indicating a notable prevalence of *JAK2* V617F and *CALR* mutations within a large cohort, despite most cases not meeting MPN diagnostic criteria [59]. The progression from *JAK2* V617F CHIP to MPN, particularly when VAF exceeds 2% or shows an increase in subsequent evaluations, indicates a significant potential for developing MPN. Longitudinal studies suggest that the transition from CHIP to MPN can span 5 to 15 years [60], with the mutation event predating the MPN diagnosis by decades, emphasizing the latent clinical emergence of the mutation. The higher occurrence rate of *JAK2* V617F CHIP compared to *CALR* mutations may reflect different clonal expansion dynamics or immune evasion mechanisms [12,61].

## 4. Germline Variants in MPN Pathogenesis

Research has uncovered the critical role of certain germline single-nucleotide polymorphisms (SNPs) and haplotypes in predisposing individuals to MPNs. Notably, the *JAK2* 46/1 or GGCC haplotype significantly increases MPN risk by two to six times [62,63,64]. This haplotype, along with SNPs in genes like *TERT*, *MECOM*, and *CHEK2* [65], though to a lesser extent, elevates MPN susceptibility through interactions with somatic mutations, promoting the expansion of the mutant hematopoietic stem cell (HSC) population. The presence of these germline variations primarily predisposes individuals to the *JAK2* V617F mutation, a common driver in MPN pathogenesis, albeit *CALR* or *MPL* mutations are also observed [66]. 

## 5. Variant Allele Frequencies in MPNs

In the realm of MPNs, VAFs have emerged as a crucial tool for delving into molecular intricacies and guiding clinical decisions. VAFs quantify the proportion of cells harboring specific mutations, such as *JAK2*, *CALR*, and *MPL*, relative to the total cell population, shedding light on clonal evolution, prognostic categorization, and therapeutic strategies in MPNs. Elevated VAFs, particularly for *JAK2* V617F, are linked to increased hematologic abnormalities and thrombotic risk in PV and ET patients, whereas *CALR* mutations correlate with distinct disease manifestations and outcomes. A higher VAF signifies more aggressive disease characteristics and poorer prognoses, especially with *CALR* mutations, emphasizing the prognostic value of VAFs in MPNs [67,68].

The role of VAFs extends beyond diagnostic implications to include prognostication and therapeutic monitoring. Higher *JAK2* V617F VAFs predict increased thrombotic risk in PV and worse survival in PMF, while *CALR* mutations generally denote a better prognosis [68]. Furthermore, mutations like *ASXL1* and *TET2* with high VAFs indicate advanced disease and increased leukemic transformation risk [69]. The dynamic nature of VAFs also aids in assessing treatment efficacy, with therapies like pegylated interferons showing potential to reduce VAFs and achieve molecular responses, a benchmark not readily achieved with other treatments [67,68,70,71].

The elucidation of VAFs within the context of MPNs epitomizes a critical advancement in our pathophysiological comprehension, heralding a new era of molecular hematopathology. 

Long-term treatment with JAK2 inhibitors such as fedratinib or ruxolitinib has been observed to lower *JAK2* V617F VAFs in myelofibrosis (MF) patients, correlating with clinical improvements like reduced spleen size and decreased recurrence post-transplant [72,73]. However, the prognostic significance of reducing *JAK2* V617F VAFs remains uncertain, with current guidelines not mandating VAF assessment for therapeutic decision-making [74]. Similarly, *CALR* mutations, particularly Type 1/Type 1-like, are associated with favorable outcomes and a lower incidence of leukemic transformation, contrasting with the poorer survival linked to Type 2/Type 2-like mutations or *JAK2* V617F mutations [68].

## 6. Genetic Variant Classifications in MPNs

In the domain of MPNs, unraveling the complexities of genetic variants poses a considerable challenge, further compounded by the need to assimilate ongoing clinical and functional research findings. Distinguishing pathogenic mutations that drive MPN pathogenesis from those associated with CHIP, which share genetic similarities but differ markedly in clinical outcomes, is particularly daunting. The identification of novel missense variants and the differentiation between somatic mutations and germline variants add layers of complexity to variant interpretation. An integrated methodology, incorporating functional studies and a comprehensive understanding of gene function and disease mechanisms, is essential for the accurate classification of variants and the prevention of misclassifying benign variants as pathogenic. The presence of VAFs around 50% might suggest a germline origin [75,76,77], yet this indicator is not always definitive, emphasizing the nuanced interpretation required.

To navigate the intricacies of somatic variant classification in malignancies, a tier-based framework has been devised by the Association for Molecular Pathology (AMP), the American Society of Clinical Oncology (ASCO), and the College of American Pathologists (CAP) [78]. This framework categorizes genetic variants into four tiers based on clinical relevance, from variants of strong clinical significance (Tier I) to benign or likely benign variants (Tier IV). This system aids in the consistent interpretation of variants, facilitating clear communication among healthcare professionals. However, interpreting Tier III variants, particularly those potentially of germline origin, remains challenging, highlighting the necessity for continuous research to refine genetic variant classification.

Recent validations attest to the efficacy of the AMP/ASCO/CAP guidelines in identifying clinically significant variants, underscoring their utility in clinical decision-making. Yet, the categorization of variants of uncertain significance calls for more defined criteria and broader databases for accurate interpretation [10]. The Clinical Genome Resource (ClinGen) Somatic Cancer Clinical Domain Working Group, the Cancer Genomics Consortium (CGC), and the Variant Interpretation for Cancer Consortium (VICC) have contributed to this by developing a Standard Operating Procedure (SOP) for classifying the oncogenicity of somatic variants. Validated against 94 somatic variants across 10 common cancer-related genes, this SOP aims to standardize somatic variant classification, categorizing evidence of oncogenicity into five categories—Oncogenic, Likely Oncogenic, Variant of Uncertain Significance (VUS), Likely Benign, and Benign—thus improving the clarity and consistency of somatic variant reporting in clinical settings [79]. This ongoing evolution in variant classification aims to refine diagnostic accuracy and foster personalized treatment strategies, with the recognition that the impact of specific mutations can vary significantly in terms of gene function and clinical outcomes, as illustrated by research on *TP53* mutations, suggesting potential complexities across different variants in the same genes [80,81]. Table 2 outlines the oncogenic/likely oncogenic variants reported in hematologic malignancies.

## 7. Chromosomal Aberrations in MPNs

The pathogenesis of MPNs, including PV, ET, and PMF, is closely linked to genetic and chromosomal aberrations, which significantly impact clinical outcomes and the disease’s evolution [43]. Studies, including one involving 649 patients with PMF, post-PV MF, or post-ET MF, have identified high-risk karyotypes as critical independent prognostic factors for the transformation to acute myeloid leukemia (AML) [82]. A retrospective analysis of cytogenetic data from 376 patients further established a strong association between abnormal karyotypes and heightened risk as defined by the MYSEC-PM model, especially for those with monosomal karyotypes, regardless of MYSEC-PM classification [83].

Karyotype classifications delineate an “unfavorable karyotype” as any abnormal karyotype excluding normal, sole anomalies of 20q-, 13q-, +9, chromosome 1 translocation/duplication, or -Y, and other sex chromosome abnormalities except -Y. In contrast, a “very high-risk (VHR) karyotype” encompasses single or multiple abnormalities of −7, i(17q), inv(3)/3q21, 12p−/12p11.2, 11q−/11q23, or other autosomal trisomies not including +8/+9, such as +21 or +19, highlighting the intricate relationship between chromosomal abnormalities and the prognosis of MPNs [6,84].

The traditional method of karyotyping, while foundational in detecting significant chromosomal alterations, faces limitations in resolution, prompting the need for more advanced techniques for a thorough analysis. The integration of NGS technologies has significantly enhanced the ability to detect chromosomal abnormalities and rearrangements, depending on the assay’s design [85]. Additionally, the use of single-nucleotide polymorphism (SNP) array methodologies and Optical Genome Mapping (OGM) has expanded the scope of diagnostic tools, allowing for the identification of structural variations and providing a more comprehensive understanding of the genetic complexities underlying MPNs [86,87,88].

## 8. Implications for Treatment: A Genetic Perspective

The treatment approach for MPNs, encompassing PV, ET, and PMF, has significantly evolved due to a deeper understanding of their genetic foundations. Discoveries of mutations in the *JAK2*, *CALR*, and *MPL* genes have refined diagnostic precision and facilitated the advent of targeted therapies.

The identification of the *JAK2* V617F mutation across MPN subtypes catalyzed a paradigm shift towards the use of JAK2 inhibitors like ruxolitinib and fedratinib, especially in managing PV and PMF. These treatments have proven effective in alleviating symptoms, diminishing spleen size, and improving the survival rates of patients harboring *JAK2* mutations. Despite their efficacy, the variability in patient responses and the emergence of resistance highlight the necessity for personalized treatment plans and the exploration of combination therapies to ensure optimal results. The ability of cells to continue proliferating despite the blockage of JAK2 signaling by inhibitors points to a complex adaptive mechanism. Although mutations such as Y931C, G993A, L884P, G935R, R938L, E864K, I960V, and E985K in the *JAK2* gene are theoretically linked to resistance [89,90,91], their near absence in clinical settings [92] indicates they are not a typical cause for resistance to JAK2 inhibitors.

*CALR* mutations, predominantly found in ET and PMF patients lacking *JAK2* mutations, have opened new therapeutic avenues beyond JAK inhibition [11]. While direct CALR-targeted treatments are in development, the existence of *CALR* mutations already informs therapeutic choices, suggesting a potentially lower thrombotic risk and influencing the selection of supportive care measures [6].

Though *MPL* mutations are less common, they are implicated in ET and PMF pathogenesis through their role in thrombopoietin signaling. The exploration of targeted treatments for *MPL* mutations, including thrombopoietin receptor agonists, underscores the ongoing expansion of therapeutic options guided by molecular diagnostics [93,94]. 

The integration of comprehensive genetic profiling into treatment decisions marks a significant advance in MPN management. This strategy allows for the identification of primary driver mutations and secondary genetic alterations, influencing disease progression, prognosis, and treatment efficacy. For example, the detection of HMR mutations may prompt consideration of more aggressive treatments, including stem cell transplantation for PMF patients [6]. Furthermore, understanding the sequential occurrence of mutations offers insights into disease dynamics and potential therapeutic interventions [41].

The management of MPN-NOS/MPN-U presents significant challenges due to the heterogeneity of its genetic and clinical presentations. Unlike other well-defined MPNs, MPN-NOS lacks standardized treatment guidelines and the utility of mutational and cytogenetic analysis as a prognostic tool in MPN-NOS/MPN-U is not well established. This limitation is due to the small number of studies, which requires further validation [95]. Treatment strategies, typically adapted from other myeloproliferative disorders, focus on symptom management and disease monitoring, often involving hydroxyurea for cytoreduction and low-dose aspirin for thromboprophylaxis based on individual risk. The potential evolution into more aggressive states requires close monitoring and, for certain eligible patients, may require consideration of allogeneic hematopoietic stem cell transplantation, although the supporting data are limited due to the rarity of this condition [95]. A comprehensive exploration of therapeutic options, however, is beyond the scope of this review.

Looking forward, the future of MPN treatment lies in precision medicine, with ongoing research into mutation-specific disease phenotypes and resistance mechanisms paving the way for next-generation therapies and combination treatments [11,96]. The implementation of real-time molecular monitoring will facilitate dynamic treatment adjustments, aligning therapeutic strategies with each patient’s evolving genetic landscape. This approach aims to not only enhance survival and quality of life for MPN patients but also set the stage for curative strategies in the future.

## 9. Emerging Technologies and Approaches

The integration of advanced genomic technologies, including whole-genome, whole-exome, and targeted sequencing, as well as single-cell genomics and comprehensive genomic profiling, has revolutionized the diagnosis and treatment of MPNs. These methodologies have enabled a detailed exploration of the genetic and epigenetic nuances of MPNs, uncovering novel mutations and fostering the development of personalized therapeutic strategies by elucidating the genetic intricacies that underpin disease heterogeneity, prognosis, and therapeutic response. Particularly, NGS and single-cell sequencing (SCS) technologies have been pivotal in revealing cellular heterogeneity, clonal architecture, and mutation acquisition order, thereby offering fresh insights into disease pathogenesis and identifying new targets for therapeutic intervention.

Integrative genomic profiling, which combines genomic, epigenomic, and transcriptomic data, provides a comprehensive view of MPN biology, revealing the functional implications of mutations and identifying biomarkers critical for monitoring disease progression and predicting therapy responsiveness. This comprehensive analysis supports the development of more effective, tailored treatments. Additionally, functional analyses of novel mutations, facilitated by CRISPR-Cas9 gene editing and single-cell functional assays, are instrumental in delineating the roles of these mutations in disease mechanisms and highlighting opportunities for targeted therapy.

The endeavor to personalize treatment strategies in MPNs involves adapting therapies to the distinct genetic and molecular landscapes of patients’ conditions (Table 3). This requires a broader molecular comprehension of MPNs, the identification of predictive biomarkers, and the structuring of clinical trials to test targeted treatments in genetically defined patient groups. The exploration of novel pharmacological interventions, gene editing techniques, RNA-based therapies, and immunotherapies is expected to significantly propel forward the care and management of MPN patients [97,98,99,100]. 

Furthermore, the shift toward comprehensive molecular classifications of MPNs represents a departure from traditional diagnostic criteria towards a framework informed by a deep understanding of genetic and epigenetic modifications. This transition aims to enhance diagnostic accuracy, enable personalized clinical approaches, and improve patient outcomes through the development of subtype-specific therapies [11,96]. Realizing this vision necessitates reducing the costs and expanding the accessibility of genomic analyses, supported by advancements in bioinformatics, data standardization, and the formulation of new clinical guidelines emphasizing molecular diagnostics. This shift heralds a new era of precision medicine for patients with MPNs, marked by a concerted effort to tailor healthcare strategies to the unique molecular signatures of individual diseases.

## 10. Discussion

The revision of diagnostic criteria for MPNs as per the latest updates from the WHO and the ICC has not seen substantial changes [3,4]. However, there has been a considerable enhancement in the molecular understanding of these diseases. The classification of MPN-NOS/MPN-U continues to present challenges, reflecting the complexities involved in diagnosing atypical cases of MPNs and underlining the necessity for further research.

The significance of mutations in *JAK2*, *CALR*, and *MPL*, along with clonal driver mutations, is pivotal in elucidating the dynamic clonal evolution observed within MPNs. This evolution contributes to disease progression, treatment resistance, and variability in prognosis. Such genetic diversity demands the development of personalized therapeutic strategies, which have been facilitated by advancements in genomic technologies, including NGS.

Prognostic models, such as the MIPSS and the GIPSS [42,43,44,45], play a crucial role in refining risk assessment and guiding treatment decisions. These models amalgamate genetic data with clinical parameters to enhance the accuracy of predicting disease outcomes, thereby improving patient management strategies. Despite their proven efficacy, the application of these models in cases of MPN-NOS/MPN-U remains to be fully validated, necessitating additional research to establish their utility in these atypical presentations [95].

Nevertheless, the interpretation of genetic variants, particularly those of uncertain significance, remains a considerable challenge. This issue underscores the need for standardized, comprehensive criteria and databases to enhance diagnostic precision and therapeutic efficacy. Efforts by entities like the ClinGen Somatic Cancer Clinical Domain Working Group and similar consortia have been instrumental in developing guidelines to tackle these challenges [79]. The establishment of consensus databases for the interpretation of these variants in hematologic malignancies, analogous to ClinVar, would benefit from the peer review process to validate the accuracy of interpretations submitted. Such collaborative efforts are essential for advancing the reliability and application of genetic data in clinical settings.

Looking to the future, MPN treatment and research are increasingly focused on gaining a deeper understanding of clonal architecture and its clinical implications. The adoption of precision medicine approaches, propelled by rapid advancements in technologies such as single-cell genomics—which, although not yet clinically utilized—promises to revolutionize personalized management. This approach is poised to enable the formulation of treatment protocols that are tailored not only to the initial presentation of the disease but also to its genetic alterations over time.

## 11. Conclusions

Recent advancements in molecular insights into MPNs signify the dawn of a new epoch in precision medicine for hematologic malignancies. The identification of mutations in *JAK2*, *CALR*, and *MPL* genes, augmented by developments in NGS, single-cell genomics, and comprehensive genomic profiling, has heralded a transformative shift in the diagnostic, prognostic, and therapeutic landscape of MPNs. The intricate genetic and epigenetic architecture of MPNs necessitates a refined, personalized approach to patient care, underscoring the critical role of precision medicine. The ongoing incorporation of advanced genomic technologies into routine clinical workflows promises to deepen our understanding of MPN pathogenesis, pave the way for novel therapeutic avenues, and enhance patient outcomes. This narrative reflects the current trajectory of MPN research and projects future efforts focused on harnessing molecular diagnostics and targeted treatment modalities to revolutionize patient management in this complex disease spectrum.

## Figures and Tables

**Table 1 cancers-16-01679-t001:** WHO 2022 and ICC diagnostic criteria for MPN [3,4].

Disease	WHO Criteria	Diagnosis Requirements: WHO	ICC 2022 Classification
PV *	Major: 1. Elevated hemoglobin (>16.5 g/dL in men, >16.0 g/dL in women) or hematocrit (>49% in men, >48% in women). 2. Bone marrow biopsy showing trilineage proliferation (panmyelosis) with pleomorphic, mature megakaryocytes. 3. Presence of *JAK2* V617F or exon 12 mutation. Minor: Subnormal serum erythropoietin level.	Option 1: All three criteria must be met. Option 2: The first two major criteria, plus the minor criterion.	Diagnostic Criteria: Option 1: All three criteria must be met. Option 2: First and third major criteria, plus the minor criterion. Note: Increased red blood cell mass is included in the diagnostic criteria.
Post-PV myelofibrosis	Major Criteria: 1. Established diagnosis of PV. 2. Bone marrow fibrosis (Grade 2 or 3). Additional Criteria: 1. Anemia (below reference range considering age, sex, altitude, or sustained loss of phlebotomy or cytoreductive treatment requirement). 2. Leukoerythroblastosis. 3. Increasing splenomegaly (increase in palpable splenomegaly >50 mm from baseline or development of newly palpable splenomegaly). 4. Development of at least two of the following symptoms: weight loss (>10% in 6 months), night sweats, unexplained fever (>37.5 °C).	Major criteria plus two additional criteria.	No significant differences noted
ET	Major Criteria: 1. Platelet count ≥ 450 × 10^9^/L. 2. Bone marrow biopsy showing: Proliferation mainly of the megakaryocytic lineage. Increased numbers of enlarged, mature megakaryocytes with hyperlobulated nuclei. No significant increase or left shift in neutrophil granulopoiesis or erythropoiesis. Very rarely, a minor (grade 1) increase in reticulin fibers. 3. Absence of WHO criteria for: *BCR::ABL1*-positive CML. PV. PMF. Other myeloid neoplasms. 4. Mutation in *JAK2*, *CALR*, or *MPL* genes. Minor Criteria: 1. Presence of a clonal marker. 2. Exclusion of reactive thrombocytosis.	Option 1: All major criteria must be met. Option 2: First three major criteria plus one minor criterion.	No significant differences noted
Post-ET myelofibrosis	Required Criteria: 1. Previous diagnosis of WHO-defined ET. 2. Bone marrow fibrosis grade 2–3 (on a scale of 0–3). Additional Criteria: 1. Anemia (below reference range for age, sex, and altitude) with >2 g/dL decrease from baseline hemoglobin. 2. Leukoerythroblastosis. 3. Increasing splenomegaly, defined as: Increase in palpable splenomegaly >50 mm from baseline. Development of newly palpable splenomegaly. 4. Elevated LDH level (above reference range). 5. Development of any two (or all three) of the following symptoms: 10% weight loss in 6 months. Night sweats. Unexplained fever (>37.5 °C).	Option 1: All required criteria must be met. Option 2: At least two additional criteria.	No significant differences noted
PMF, Prefibrotic Stage	Major Criteria: 1. Megakaryocytic proliferation and atypia (without reticulin fibrosis grade > 1), accompanied by: Increased age-adjusted bone marrow cellularity. Granulocytic proliferation. Often decreased erythropoiesis. 2. Absence of diagnostic criteria for: CML. PV. ET. MDS. Other defined myeloid neoplasms. 3. Presence of *JAK2*, *CALR*, or *MPL* mutation, another clonal marker, or absence of reactive bone marrow fibrosis. Minor Criteria: 1. Anemia not attributed to a comorbid condition. 2. Leukocytosis ≥ 11 × 10^9^/L. 3. Clinically and/or imaging-detected splenomegaly. 4. LDH level above the upper limit of the institutional reference range.	Required: All three major criteria. Additional: At least one minor criterion. Confirmation: Minor criteria must be confirmed in two consecutive determinations.	No significant differences noted
PMF, Fibrotic Stage	Major Criteria: 1. Megakaryocytic proliferation and atypia, with reticulin and/or collagen fibrosis grade 2 or 3. 2. Does not meet diagnostic criteria for: CML. PV. ET. MDS. Other defined myeloid neoplasms. Presence of *JAK2*, *CALR*, or *MPL* mutation, another clonal marker, or absence of reactive bone marrow fibrosis. Minor Criteria: 1. Anemia not attributed to a comorbid condition. 2. Leukocytosis ≥ 11 × 10^9^/L. 3. Clinically and/or imaging-detected splenomegaly. 4. LDH level above the upper limit of the institutional reference range. 5. Leukoerythroblastosis.	Required: All three major criteria. Additional: At least one minor criterion. Confirmation: Minor criteria must be confirmed in two consecutive determinations.	No significant differences noted
MPN-NOS/MPN-U	Required Criteria: 1. Presence of any one of the following features: Clinical and hematological features of an MPN (e.g., splenomegaly, leukocytosis, thrombocytosis) without significant monocytosis and eosinophilia. Bone marrow hypercellularity with megakaryocytic hyperplasia and varying degrees of granulocytic and erythroid hyperplasia, without dysplastic features. Clinical and morphological features may be discrepant. 2. Does not meet criteria for: Other MPNs. MDS. MDS/MPN overlap syndromes. Myeloid/lymphoid neoplasms with eosinophilia and tyrosine kinase gene fusions. 3. Presence of driver mutations such as *JAK2*, *CALR*, or *MPL* mutations, or another clonal marker. Exclusion Criteria: 1. Insufficient clinical data or inadequate bone marrow specimen for accurate evaluation and classification. 2. Recent history of cytotoxic or growth factor therapy, especially when dysplastic features are present.	Required: Presence of all required criteria and absence of all exclusion criteria	Similar diagnostic criteria

* Abbreviations: PV: Polycythemia Vera; ET: Essential Thrombocythemia; CML: Chronic Myeloid Leukemia; PMF: Primary Myelofibrosis; MDS: Myelodysplastic Neoplasm; MPN-NOS: Myeloproliferative Neoplasm, Not Otherwise Specified; MPN-U: Myeloproliferative Neoplasm, Unclassifiable; WHO: World Health Organization; ICC: International Consensus Classification.

**Table 2 cancers-16-01679-t002:** Mutations in MPNs and their clinical implications.

Gene	Frequency in MPN [12]	Reported Oncogenic/Likely Oncogenic Mutations and (Reference Transcripts) [3]	Significance and Impact on Prognosis [6]
Disease Drivers			
*JAK2*	PV *: 98% (~95% V617, ~4% exon 12) ET: 55% PMF: 60%	V617F; Missense/indel in aa range: pp. 536–547 (NM_004972)	WHO/ICC criterion for diagnosis; intermediate prognosis with a heightened risk of thrombosis relative to *CALR* type 1 mutation carriers [6,13]
*MPL*	PV: 0% ET: 5–7%, PMF 7–10%	S505G, S505N, S505C, L510P, del513, W515A, W515R, W515K, W515S, W515L, A519T, A519V, Y591D, W515-518KT. (NM_005373)	WHO/ICC criterion for diagnosis; intermediate prognosis with a heightened risk of thrombosis relative to *CALR* type 1 mutation carriers [6,13]
*CALR*	PV: 0% ET: 25–30% PMF: 20–30%	Frameshift in exon 9 (NM_004343)	WHO/ICC criterion for diagnosis; *CALR* 1: enhanced OS and reduced thrombosis risk in comparison to those with *JAK2* mutations and TN-PMF, as well as better OS than *CALR* type 2 mutation carriers [14,15,16]; *CALR* 2: lower OS than *CALR* 1 [17]
Clonal Drivers			
*DNMT3A*	PV: 5–10% ET: 1–5% PMF: 8–12%	Frameshift/nonsense/splice-site; missense in aa range: pp. 292–350, 482–614, 634–912 (NM_022552)	Inferior OS post-HCT [18]
*IDH1*	PV: 1–2% ET: 1–2% PMF: 5–6%	Frameshift/nonsense/splice-site in exon 11–12 (NM_015338)	HMR Inferior OS and reduced PFS post-HCT [8]
*IDH2*	PV: 1–2% ET: 1–2% PMF: 5–6%	Missense at R132 (NM_005896)	HMR Inferior OS and reduced PFS post-HCT [8]
*ASXL1*	PV: 2–7% ET: 5–10% PMF: 15–35%	Frameshift/nonsense/splice-site in exon 11–12 (NM_015338)	HMR Adverse impact, particularly in PMF; marked by poorer OS and LFS, including post-HCT [8]
*EZH2*	PV: 1–2% ET: 1–2% PMF: 7–10%	Frameshift/nonsense/splice-site; missense in SET domain (pp. 617–732) (NM_001203247)	HMR Inferior OS [8]
*NRAS*	PV: <2% ET: <2% PMF: 2–4%	Missense at G12/G13/Q61 (NM_002524)	Inferior OS [19]
*KRAS*	PV: <2% ET: <2% PMF: 2%	Missense at G12/G13/Q61 (NM_033360)	Similar to *NRAS*
*CBL*	PV: <2% ET: <2% PMF: 4%	Missense in Linker/RING finger domains (pp. 345–434) (NM_005188)	Inferior OS post-HCT [18]
*SRSF2*	PV: <2% ET: <2% PMF: 6–14%	Missense/in-frame deletion involving P95 (NM_003016)	HMR in all MPNs Inferior OS and LFS; adverse prognosis in transformation [8]
*U2AF1*	PV: <2% ET: <2% PMF: 7–10%	Missense at S34/Q157 (NM_006758)	HMR Adverse prognosis in PMF and secondary AML; diminished OS post-HCT, with *U2AF1* Q157 mutation associated with worse outcomes compared to *U2AF1* S34 mutations or unmutated MF [18]
*TP53*	PV: <2% ET: <2% PMF: 2–5% Increased frequency in advanced stages/post-MPN AML	Frameshift/nonsense/splice-site; missense in aa range: pp. 72, 95–288, 337 (NM_001126112)	Higher likelihood of leukemic transformation [20]
*TET2*	PV: 10–20% ET: 3–10% PMF: 10–20%	Frameshift/nonsense/splice-site; aa range: pp. 1104–1481, 1843–2002 (NM_001127208)	No consensus impact on prognosis
*SH2B3 (LNK)*	PV: 2–9% ET: 1–3% PMF: 2–4%	Frameshift/nonsense/splice-site; Missense at E208Q (NM_005475) [21]	Reported as potential driver in *JAK2* negative MPN [22]
*RUNX1*	PV: <2% ET: <2% PMF: 2–3%	Frameshift/nonsense/splice-site, S73F, H78Q, H78L, R80C, R80P, R80H, L85Q, P86L, P86H, S114L, D133Y, L134P, R135G, R135K, R135S, R139Q, R142S, A165V, R174Q, R177L, R177Q, A224T, D171G, D171V, D171N, R205W, R223C (NM_001001890)	Frequent in leukemic transformation [23,24]
*SF3B1*	PV: 2–3% ET: 2–5% PMF: 5–7% The possibility of mixed myelodysplastic component should be considered [3,4]	Missense in terminal HEAT domains (pp. 529–1201) (NM_012433)	Adverse impact in ET [12]

* Abbreviations: PV: Polycythemia Vera; ET: Essential Thrombocythemia; PMF: Primary Myelofibrosis; WHO: World Health Organization; ICC: International Consensus Classification; OS: overall survival; LFS: leukemia-free survival; HCT: hematopoietic cell transplantation; HMR: high-molecular-risk mutation; PFS: progression-free survival.

**Table 3 cancers-16-01679-t003:** Personalized treatment strategies for MPN based on molecular changes [11,96].

Mutation	Targeted Therapy Options	Clinical Trial Evidence	FDA Approval Status
*JAK2* V617F	Ruxolitinib, Pacritinib, Fedratinib	Ruxolitinib showed significant spleen volume reduction and improved quality of life in MF and PV patients	Ruxolitinib, Pacritinib, and Fedratinib are FDA-approved
*CALR* Mutations	Immunological therapies, CALR-targeted, mutant *CALR* peptide vaccine	Investigations on disrupting CALRdel52-MPL signaling complexes in *CALR*-mutated cells	No specific FDA approvals for CALR-targeted therapies yet
Telomerase Activity	Imetelstat (Telomerase inhibitor)	A phase 2 trial showed clinical improvements in intermediate-2/high-risk MF patients relapsed or refractory to ruxolitinib	Not yet FDA-approved; phase 3 trial ongoing
HSP90 *	PU-H71 (HSP90 inhibitor)	Early phase clinical trials ongoing for safety, tolerability, and pharmacokinetic profile in MPN patients	Not yet FDA-approved
MDM2/TP53 Pathway	Idasanutlin, KRT232 (MDM2 antagonists)	Phase I/Ib study in AML patients with idasanutlin showed durable responses; ongoing studies in MF. Idasanutlin in PV showed rapidly reduced *JAK2* allele burden in PV patients [101]	Not yet FDA-approved for MPNs
Hepcidin Mimetics in PV	Rusfertide (PTG-300)	Phase 2 trials showed reduced hematocrit levels and therapeutic phlebotomy needs in PV patients	Phase 3 trial underway; not yet FDA-approved
Bcl-2/Bcl-xL Inhibition	Navitoclax	Phase 2 trial showed safety and efficacy in MF patients, with ongoing phase 3 trials	Not yet FDA-approved; phase 3 trial ongoing
Interferons	Pegylated interferons	Induce durable molecular responses and preferentially deplete *JAK2*-mutated HSCs, showing efficacy in ET and PV	Used in clinical practice but specific FDA approval varies
CD123 Targeted Therapy	Tagraxofusp (SL-401)	Phase I/II clinical trial ongoing in intermediate- or high-risk and relapsed/refractory MF patients	FDA-approved for BPDCN, not specifically for MPNs

* Abbreviations: HSP90: Heat shock protein-90; BPDCN: Blastic plasmacytoid dendritic cell neoplasm.

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
