# Peer review of "Advances in Molecular Understanding of Polycythemia Vera, Essential Thrombocythemia, and Primary Myelofibrosis: Towards Precision Medicine"

_cancers, 2024, doi:10.3390/cancers16091679_

Round 1

Reviewer 1 Report

Comments and Suggestions for Authors

The authors provided a comprehensive review of the current literature on molecular diagnosis and targeted therapies for myeloproliferative neoplasms (MPNs).  While there is a vast amount of information available on the molecular diagnosis of MPNs, there are few articles that summarize targeted treatment approaches. From this perspective, the review will certainly be of interest to a diverse range of readers. I would suggest addressing few minor points to make it even more attractive for the readers. 

1 Please discuss the peculiarities of diagnosis and treatment for myeloproliferative neoplasms not otherwise specified (NOS), in addition to PV, ET and PMF.

2 Is should be beneficial to include table with current WHO diagnostic criteria for PV, ET, PMF and MPN-NOS.

3 Human gene names should be written in italic capital letters. At the moment, inconsistent typography is used throughout the article.

4 In columns 3 and 4 of the table 1, the data for rows 1 and 2 are mixed up and/or incorrect.

5 In table 2, V617F is marked with an asterisk, but there is no corresponding footnote provided.

Author Response

Thank you for the positive and insightful feedback on our submission. We have integrated your valuable suggestions and are eager to hear any additional comments you might have. Below are our responses to your remarks.

1 Please discuss the peculiarities of diagnosis and treatment for myeloproliferative neoplasms not otherwise specified (NOS), in addition to PV, ET and PMF.

We appreciate the suggestion to elaborate on MPN-NOS. We have incorporated a new section, numbered Section 2, in the manuscript that specifically addresses the diagnostic criteria for PV, ET, PMF, and MPN-NOS, emphasizing the diagnostic complexities associated with MPN-NOS. Furthermore, we have expanded the discussion on the unique treatment challenges of MPN-NOS in the existing Section 8. We have noted that a detailed discussion of therapy options is outside the scope of this review.

2 Is should be beneficial to include table with current WHO diagnostic criteria for PV, ET, PMF and MPN-NOS.

We agree that including a table summarizing the current WHO diagnostic criteria for PV, ET, PMF, and MPN-NOS would enhance the manuscript's utility. Table 1 has been created to provide this overview.

3 Human gene names should be written in italic capital letters. At the moment, inconsistent typography is used throughout the article.

Thank you for pointing out the inconsistency in the typography of human gene names. The manuscript has been revised to ensure all gene names are consistently presented in italic capital letters, adhering to the standard formatting.

4 In columns 3 and 4 of the table 1, the data for rows 1 and 2 are mixed up and/or incorrect.

We apologize for the errors in Table 1 (now Table 2) and appreciate your careful reading. The data in columns 3 and 4 for rows 1 and 2 have been corrected.

5 In table 2, V617F is marked with an asterisk, but there is no corresponding footnote provided.

The asterisk next to V617F in Table 2 (now Table 3) was mistakenly included without an accompanying footnote. This error has been corrected, and the relevant footnote has now been added.

Reviewer 2 Report

Comments and Suggestions for Authors

Tashkandi's article is an interesting review of the latest advances in the molecular understanding of myeloproliferative neoplasms, focusing on molecular diagnostics and targeted treatment modalities, in order to revolutionize patient management.

I appreciated figure 1, but the small writing makes it difficult to read and interpret. Since it is the only image, I recommend paying more attention to the graphics.

Author Response

Thank you for the positive and insightful feedback on our submission. We have integrated your valuable suggestions and are eager to hear any additional comments you might have. Below are our responses to your remarks.

I appreciated figure 1, but the small writing makes it difficult to read and interpret. Since it is the only image, I recommend paying more attention to the graphics.

We appreciate your feedback regarding the legibility of Figure 1. This figure has been revised to enhance its readability, with adjustments made to the font size and layout.

Reviewer 3 Report

Comments and Suggestions for Authors

- The authors provide a comprehensive review article on the molecular landscape of PV, ET, and PMF, highlighting the diagnostic, prognostic, and therapeutic implications of these genetic findings. The authors dealed with the challenges of diagnosing and treating patients with prognostic mutations, the clonal evolution, and the impact of emerging technologies like next-generation sequencing and single-cell genomics on the field.

- The article is timing and novel and clearly fills a gap in the current literature.

- Overall considered, all the sections of the article are well written but there is not a discussion section. The authors must add a discussion section in which they should highlight the authors' critical point of view and make comparisons with other studies in the research field.

- English language is fine.

Author Response

Thank you for your positive comments. We are pleased to hear that the sections are well-received. Below are our responses to your remarks.

- Overall considered, all the sections of the article are well written but there is not a discussion section. The authors must add a discussion section in which they should highlight the authors' critical point of view and make comparisons with other studies in the research field.

Thank you for your constructive feedback regarding the addition of a discussion section to our manuscript. We have incorporated a discussion section as you suggested.